# Evolutionary Reinforcement Learning for Sample-Efficient Multiagent Coordination

## Abstract

Many cooperative multiagent reinforcement learning environments provide agents with a sparse team-based reward, as well as a dense agent-specific reward that incentivizes learning basic skills. Training policies solely on the team-based reward is often difficult due to its sparsity. Also, relying solely on the agent-specific reward is sub-optimal because it usually does not capture the team coordination objective. A common approach is to use reward shaping to construct a proxy reward by combining the individual rewards. However, this requires manual tuning for each environment. We introduce Multiagent Evolutionary Reinforcement Learning (MERL), a split-level training platform that handles the two objectives separately through two optimization processes. An evolutionary algorithm maximizes the sparse team-based objective through neuroevolution on a population of teams. Concurrently, a gradient-based optimizer trains policies to only maximize the dense agent-specific rewards. The gradient-based policies are periodically added to the evolutionary population as a way of information transfer between the two optimization processes. This enables the evolutionary algorithm to use skills learned via the agent-specific rewards toward optimizing the global objective. Results demonstrate that MERL significantly outperforms state-of-the-art methods, such as MADDPG, on a number of difficult coordination benchmarks.

## 1 Introduction

Cooperative multiagent reinforcement learning (MARL) studies how multiple agents can learn to coordinate as a team toward maximizing a global objective. Cooperative MARL has been applied to many real world applications such as air traffic control (Tumer and Agogino, 2007), multi-robot coordination (Sheng et al., 2006; Yliniemi et al., 2014), communication and language (Lazaridou et al., 2016; Mordatch and Abbeel, 2018), and autonomous driving (Shalev-Shwartz et al., 2016).

Many such environments endow agents with a team reward that reflects the team's coordination objective, as well as an agent-specific local reward that rewards basic skills. For instance, in soccer, dense local rewards could capture agent-specific skills such as passing, dribbling and running. The agents must then coordinate when and where to use these skills in order to optimize the team objective, which is winning the game. Usually, the agent-specific reward is dense and easy to learn from, while the team reward is sparse and requires the cooperation of all or most agents.

Having each agent directly optimize the team reward and ignore the agent-specific reward usually fails or is sample-inefficient for complex tasks due to the sparsity of the team reward. Conversely, having each agent directly optimize the agent-specific reward also fails because it does not capture the team's objective, even with state of the art multiagent RL algorithms such as MADDPG (Lowe et al., 2017).

One solution to this problem is to use reward shaping, where extensive domain knowledge about the task is used to create a proxy reward function (Rahmattalabi et al., 2016). Constructing this proxy reward function is difficult in complex environments, and is domain-dependent. Apart from requiring domain knowledge and manual tuning, this approach also poses risks of changing the underlying problem itself (Ng et al., 1999). Simple approaches to creating a proxy reward via linear combinations of the two objectives also fail to solve or generalize to complex coordination tasks (Devlin et al., 2011; Williamson et al., 2009).

In this paper, we introduce Multiagent Evolutionary Reinforcement Learning (MERL), a state-of-the-art algorithm for cooperative MARL that does not require reward shaping. MERL is a split-level training platform that combines gradient-based and gradient-free optimization. The gradient-free optimizer is an evolutionary algorithm that maximizes the team objective through neuroevolution. The gradient-based optimizer is a policy gradient algorithm that maximizes each agent's dense, local rewards. These gradient-based policies are periodically copied into the evolutionary population. The two processes operate concurrently and share information through a shared replay buffer.

A key strength of MERL is that it is a general method which does not require domain-specific reward shaping. This is because MERL optimizes the team objective directly while simultaneously leveraging agent-specific rewards to learn basic skills. We test MERL in a number of multiagent coordination benchmarks. Results demonstrate that MERL significantly outperforms state-of-the-art methods such as MADDPG, while using the same observations and reward functions. We also demonstrate that MERL scales gracefully to increasing complexity of coordination objectives where MADDPG and its variants fail to learn entirely.

## 2 BACKGROUND AND RELATED WORK

**Markov Games:** A standard reinforcement learning (RL) setting is often formalized as a Markov Decision Process (MDP) and consists of an agent interacting with an environment over a finite number of discrete time steps. This formulation can be extended to multiagent systems in the form of partially observable Markov games (Littman, 1994; Lowe et al., 2017). An $N$-agent Markov game is defined by a global state of the world, $\mathcal{S}$, and a set of $N$ observations $\{\mathcal{O}_i\}$ and $N$ actions $\{\mathcal{A}_i\}$ corresponding to the $N$ agents. At each time step $t$, each agent observes its corresponding observation $O_i^t$ and maps it to an action $A_i^t$ using its policy $\pi_i$.

Each agent receives a scalar reward $r_i^t$ based on the global state $\mathcal{S}_t$ and joint action of the team. The world then transitions to the next state $\mathcal{S}_{t+1}$ which produces a new set of observations $\{\mathcal{O}_i\}$. The process continues until a terminal state is reached. $R_i = \sum_{t=0}^{T} \gamma^t r_i^t$ is the total return for agent $i$ with discount factor $\gamma \in (0, 1]$. Each agent aims to maximize its expected return.

**TD3:** Policy gradient (PG) methods frame the goal of maximizing the expected return as the minimization of a loss function. A widely used PG method for continuous, high-dimensional action spaces is DDPG (Lillicrap et al., 2015). Recently, (Fujimoto et al., 2018) extended DDPG to Twin Delayed DDPG (TD3), addressing its well-known overestimation problem. TD3 is the state-of-the-art, off-policy algorithm for model-free DRL in continuous action spaces.

TD3 uses an actor-critic architecture (Sutton and Barto, 1998) maintaining a deterministic policy (actor) $\pi : \mathcal{S} \rightarrow \mathcal{A}$, and two distinct critics $\mathcal{Q} : \mathcal{S} \times \mathcal{A} \rightarrow \mathbb{R}_i$. Each critic independently approximates the actor's action-value function $\mathcal{Q}^\pi$. A separate copy of the actor and critics are kept as target networks for stability and are updated periodically. A noisy version of the actor is used to explore the environment during training. The actor is trained using a noisy version of the sampled policy gradient computed by backpropagation through the combined actor-critic networks. This mitigates overfitting of the deterministic policy by smoothing the policy gradient updates.

**Evolutionary Reinforcement Learning (ERL)** is a hybrid algorithm that combines Evolutionary Algorithms (EAs) (Floreano et al., 2008; Lüders et al., 2017; Fogel, 2006; Spears et al., 1993), with policy gradient methods (Khadka and Tumer, 2018). Instead of discarding the data generated during a standard EA rollout, ERL stores this data in a central replay buffer shared with the policy gradient's own rollouts - thereby increasing the diversity of the data available for the policy gradient learners. Since the EA directly optimizes for episode-wide return, it biases exploration towards states with higher long-term returns. The policy gradient algorithm which learns using this state distribution inherits this implicit bias towards long-term optimization. Concurrently, the actor trained by the policy gradient algorithm is inserted into the evolutionary population allowing the EA to benefit from the fast gradient-based learning.

**Related Work**: Lowe et al. (2017) introduced MADDPG which tackled the inherent non-stationarity of a multiagent learning environment by leveraging a critic which had full access to the joint state and action during training. Foerster et al. (2018b) utilized a similar setup with a centralized critic across agents to tackle StarCraft micromanagement tasks. An algorithm that could explicitly model other agents' learning was investigated in Foerster et al. (2018a). However, all these approaches rely

on a dense agent reward that properly captures the team objective. Methods to solve for these types of agent-specific reward functions were investigated in Li et al. (2012) but were limited to tasks with strong simulators where tree-based planning could be used.

A closely related work to MERL is (Liu et al., 2019) where Population-Based Training (PBT) (Jaderberg et al., 2017) is used to optimize the relative importance between a collection of dense, shaped rewards automatically during training. This can be interpreted as a singular central reward function constructed by scalarizing a collection of reward signals where the scalarization coefficients are adaptively learned during training. In contrast, MERL optimizes its reward functions independently with information transfer across them facilitated through shared replay buffers and policy migration directly. This form of information transfer through a shared replay buffer has been explored extensively in recent literature (Colas et al., 2018; Khadka et al., 2019).

## 3  MULTIAGENT EVOLUTIONARY REINFORCEMENT LEARNING

MERL leverages both agent-specific and team objectives through a hybrid algorithm that combines gradient-free and gradient-based optimization. The gradient-free optimizer is an evolutionary algorithm that maximizes the team objective through neuroevolution. The gradient-based optimizer trains policies to maximize agent-specific rewards. These gradient-based policies are periodically added to the evolutionary population and participate in evolution. This enables the evolutionary algorithm to use agent-specific skills learned by training on the agent-specific rewards toward optimizing the team objective without needing to resort to reward shaping.

---

**Algorithm 1** Multiagent Evolutionary Reinforcement Learning

---

1: Initialize a population of $k$ multi-head teams $pop_\pi$, each with weights $\theta^\pi$ initialized randomly
2: Initialize a shared critic $\mathcal{Q}$ with weights $\theta^\mathcal{Q}$
3: Initialize an ensemble of $N$ empty cyclic replay buffers $\mathcal{R}^k$, one for each agent
4: Define a white Gaussian noise generator $\mathcal{W}_g$ random number generator $r() \in [0, 1)$
5: **for** generation = 1, $\infty$ **do**
6:     **for** team $\pi \in pop_\pi$ **do**
7:         $g, \mathcal{R}$ = Rollout ($\pi, \mathcal{R}$, noise=None, $\xi$)
8:         _, $\mathcal{R}$ = Rollout ($\pi, \mathcal{R}$, noise=$\mathcal{W}_g, \xi = 1$)
9:         Assign $g$ as $\pi$'s fitness
10:     **end for**
11:     Rank the population $pop_\pi$ based on fitness scores
12:     Select the first $e$ teams $\pi \in pop_\pi$ as elites
13:     Select the remaining $(k - e)$ teams $\pi$ from $pop_\pi$, to form Set $S$ using tournament selection
14:     **while** $|S| < (k - e)$ **do**
15:         Single-point crossover between a randomly sampled $\pi \in e$ and $\pi \in S$ and append to $S$
16:     **end while**
17:     **for** Agent $k$=1,$N$ **do**
18:         Randomly sample a minibatch of $T$ transitions $(o_i, a_i, l_i, o_{i+1})$ from $R^k$
19:         Compute $y_i = l_i + \gamma \min_{j=1,2} \mathcal{Q}'_j(o_{i+1}, a^\sim | \theta^{\mathcal{Q}'_j})$
20:         where $a^\sim = \pi'_{pg}(k, o_{i+1} | \theta^{\pi'_{pg}})$ [action sampled from the $k^{th}$ head of $\pi'_{pg}$] $+\epsilon$
21:         Update $\mathcal{Q}$ by minimizing the loss: $L = \frac{1}{T} \sum_i (y_i - \mathcal{Q}(o_i, a_i | \theta^\mathcal{Q})^2$
22:         Update $\pi^k_{pg}$ using the sampled policy gradient

$$\nabla_{\theta^\pi_{pg}} J \sim \frac{1}{T} \sum \nabla_a \mathcal{Q}(o, a | \theta^\mathcal{Q})|_{o=o_i, a=a_i} \nabla_{\theta^\pi_{pg}} \pi^k_{pg}(s | \theta^\pi_{pg})|_{o=o_i}$$

23:         Soft update target networks: $\theta^{\pi'} \Leftarrow \tau\theta^\pi + (1 - \tau)\theta^{\pi'}$ and $\theta^{\mathcal{Q}'} \Leftarrow \tau\theta^\mathcal{Q} + (1 - \tau)\theta^{\mathcal{Q}'}$
24:     **end for**
25:     Migrate the policy gradient team $pop_j$: for weakest $\pi \in pop^j_\pi : \theta^\pi \Leftarrow \theta^{\pi_{pg}}$
26: **end for**

---

**Policy Topology:** We represent our multiagent (*team*) policies using a multi-headed neural network $\pi$ as illustrated in Figure 1. The head $\pi^k$ represents the $k$-th agent in the team. Given an incoming observation for agent $k$, only the output of $\pi^k$ is considered as agent $k$'s response. In essence, all

agents act independently based on their own observations while sharing weights (and by extension, the features) in the lower layers (*trunk*). This is commonly used to improve learning speed (Silver et al., 2017). Further, each agent $k$ also has its own replay buffer $(R^k)$ which stores its *experience* defined by the tuple *(state, action, next state, local reward)* for each interaction with the environment (*rollout*) involving that agent.

**Team Reward Optimization:** Figure 2 illustrates the MERL algorithm. A population of multi-headed teams, each with the same topology, is initialized with random weights. The replay buffer $\mathcal{R}^k$ is shared by the $k$-th agent across all teams. The population is then *evaluated* for each rollout. The team reward for each team is disbursed at the end of the episode and is considered as its **fitness score**. A **selection** operator selects a portion of the population for survival with probability proportionate to their fitness scores. The weights of the teams in the population are probabilistically *perturbed* through mutation and crossover operators to create the next *generation* of teams. A portion of the teams with the highest relative fitness are preserved as elites. At any given time, the team with the highest fitness, or the *champion*, represents the best solution for the task.

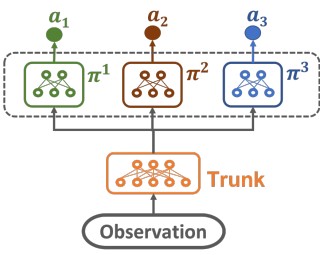

Figure 1: Team represented as multi-headed policy net $\pi$

**Policy Gradient:** The procedure described so far resembles a standard EA except that each agent $k$ stores each of its experiences in its associated replay buffer $(R^k)$ instead of just discarding it. However, unlike EA, which only learns based on the low-fidelity global reward, MERL also learns from the experiences within episodes of a rollout using policy gradients. To enable this kind of "local learning", MERL initializes one multi-headed policy network $\pi_{pg}$ and one critic $\mathcal{Q}$. A noisy version of $\pi_{pg}$ is then used to conduct its own set of rollouts in the environment, storing each agent $k$'s experiences in its corresponding buffer $(R^k)$ similar to the evolutionary rollouts.

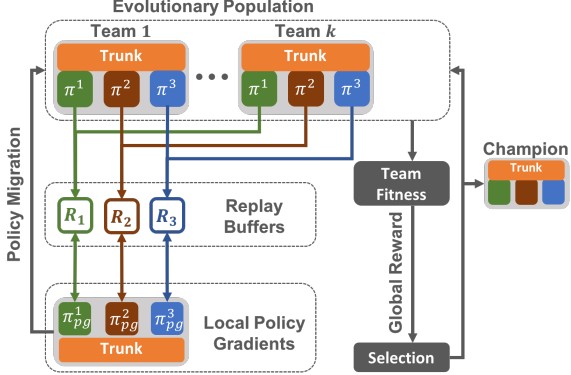

Figure 2: High level schematic of MERL highlighting the integration of local and global reward functions

**Agent-Specific Reward Optimization:** Crucially, each agent's replay buffer is kept separate from that of every other agent to ensure diversity amongst the agents. The shared critic samples a random mini-batch uniformly from each replay buffer and uses it to update its parameters using gradient descent. Each agent $\pi_{pg}^k$ then draws a mini-batch of experiences from its corresponding buffer $(R^k)$ and uses it to sample a policy gradient from the shared critic. Unlike the teams in the evolutionary population which directly seek to optimize the team reward, $\pi_{pg}$ seeks to maximize the agent-specific local reward while exploiting the experiences collected via evolution.

**Skill Migration:** Periodically, the $\pi_{pg}$ network is copied into the evolving population of teams and can propagate its features by participating in evolution. This is the core mechanism that combines policies learned via agent-specific and team rewards. Regardless of whether the two rewards are aligned, evolution ensures that only the performant derivatives of the migrated network are retained. This mechanism guarantees protection against destructive interference commonly seen when a direct scalarization between two reward functions is attempted. Further, the level of information exchange is automatically adjusted during the process of learning, in contrast to being manually tuned by an expert designer.

Algorithm 1 provides a detailed pseudo-code of the MERL algorithm. The choice of hyperparameters is explained in the Appendix. Additionally, our source code [1] is available online.

---

[1] https://tinyurl.com/y6erclts

## 4 EXPERIMENTS

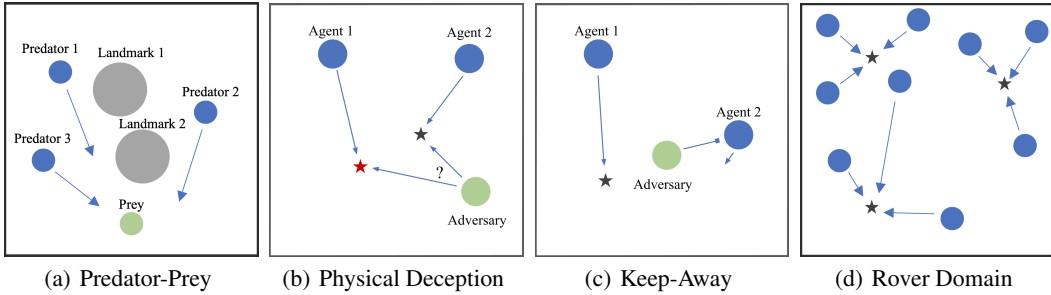

|  (a) Predator-Prey | (b) Physical Deception | (c) Keep-Away | (d) Rover Domain |

Figure 3: Illustration of environments tested (Lowe et al., 2017; Rahmattalabi et al., 2016)

We adopt environments from (Lowe et al., 2017) and (Rahmattalabi et al., 2016) to perform our experiments. Each environment consists of multiple agents and landmarks in a two-dimensional world. Agents take continuous control actions to move about the world. Figure 3 illustrates the four environments which are described in more detail below.

**Predator-Prey**: In this environment, $N$ slower cooperating agents (predators) must chase the faster adversary (prey) around an environment with $L$ large landmarks in randomly-generated locations. The predators get a reward when they catch (touch) the prey while the prey is penalized. The team reward for the predators is the cumulative number of prey-touches in an episode. Each predator can also compute the average distance to the prey and use it as its agent-specific reward. All agents observe the relative positions and velocities of the other agents as well as the positions of the landmarks. The prey can accelerate 33% faster than the predator and has a higher top speed. We tests two versions termed simple and hard predator-prey where the prey is 30% and 100% faster, respectively. Additionally, the prey itself learns dynamically during training. We use DDPG (Lillicrap et al., 2015) as a learning algorithm for training the prey policy. All of our candidate algorithms are tested on their ability to train the team of predators in catching this prey.

**Physical Deception**: $N$ agents cooperate to reach a single target Point of Interest (POI) among $N$ POIs. They are rewarded based on the closest distance of any agent to the target. A lone adversary also desires to reach the target POI. However, the adversary does not know which of the POIs is the correct one. Thus the cooperating agents must learn to spread out and cover all POIs so as to deceive the adversary as they are penalized based on the adversary's distance to the target. The team reward for the agents is then the cumulative reward in an episode. We use DDPG (Lillicrap et al., 2015) to train the adversary policy.

**Keep-Away**: In this scenario, a team of $N$ cooperating agents must reach a target POI out of $L$ total POIs. Each agent is rewarded based on its distance to the target. We construct the team reward as simply the sum of the agent-specific rewards in an episode. An adversary also has to occupy the target while keeping the cooperating agents from reaching the target by pushing them away. To incentivize this behavior, the adversary is rewarded based on its distance to the target POI and penalized based on the distance of the target from the nearest cooperating agent. Additionally, it does not know which of the POIs is the target and must infer this from the movement of the agents. DDPG (Lillicrap et al., 2015) is used to train the adversary policy.

**Rover Domain**: This environment is adapted from (Rahmattalabi et al., 2016). Here, $N$ agents must cooperate to reach a set of $K$ POIs. Multiple agents need to simultaneously go to the same POI in order to observe it. The number of agents required to observe a POI is termed the coupling requirement. Agents do not know and must infer the coupling factor from the rewards obtained. If a team with fewer agents than this number go to a POI, no reward is observed. The team's reward is the percentage of POIs observed at the end of an episode.

Each agent can also locally compute its distance to its closest POI and use it as its agent-specific reward. Its observation comprises two channels to detect POIs and rovers, respectively. Each channel receives intensity information over $10°$ resolution spanning the $360°$ around the agent's position loosely based on the characteristic of a Pioneer robot (Thrun et al., 2000). This is similar to a LIDAR. Since it returns the closest reflector, occlusions make the problem partially-observable. A coupling

factor of 1 is similar to the cooperative navigation task in Lowe et al. (2017). We test coupling factors from 1 to 7 to capture extremely complex coordination objectives.

**Compared Baselines:** We compare the performance of MERL with a standard neuroevolutionary algorithm (EA) (Fogel, 2006), MADDPG (Lowe et al., 2017) and MATD3, a variant of MADDPG that integrates the improvements described within TD3 (Fujimoto et al., 2018) over DDPG. Internally, MERL uses EA and TD3 as its team-reward and agent-specific reward optimizer, respectively. MADDPG was chosen as it is the state-of-the-art multiagent RL algorithm. We implemented MATD3 to ensure that the differences between MADDPG and MERL do not originate from having the more stable TD3 over DDPG.

**Methodology for Reported Metrics:** For MATD3 and MADDPG, the team network was periodically tested on 10 task instances without any exploratory noise. The average score was logged as its performance. For MERL and EA, the team with the highest fitness was chosen as the champion for each generation. The champion was then tested on 10 task instances, and the average score was logged. This protocol shielded the reported metrics from any bias of the population size. We conduct 5 statistically independent runs with random seeds from $\{2019, 2023\}$ and report the average with error bars showing a $95\%$ confidence interval. All scores reported are compared against the number of environment steps (frames). A step is defined as the multiagent team taking a joint action and receiving a feedback from the environment. To make the comparisons fair across single-team and population-based algorithms, all steps taken by all teams in the population are counted cumulatively.

## 5 RESULTS

**Predator-Prey:** Figure 4 shows the comparative performance in controlling the team of predators in the Predator-Prey environment. Note that this is an adversarial environment where the prey dynamically adapts against the predators. The prey (considered as part of the environment in this analysis) uses DDPG to learn constantly against our team of predators. This is why predator performance (measured as number of prey touches) exhibits ebb and flow during learning. MERL outperforms MATD3, EA, and MADDPG across both simple and hard variations of the task. EA seems to be approaching MERL's performance but is significantly slower to learn. This is an expected behavior for neuroevolutionary methods which are known to be sample-inefficient. In contrast, MERL, by virtue of its fast policy-gradient components, learns significantly faster.

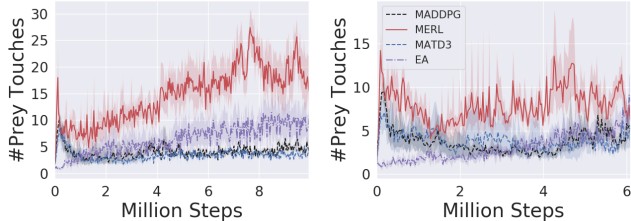

Figure 4: Performance on Predator-Prey where the prey is $30\%$ faster (left) and $100\%$ faster (right), respectively.

**Physical Deception:** Figure 5 (left) shows the comparative performance in controlling the team of agents in the Physical Deception environment. The performance here is largely based on how close the adversary comes to the target POI. Since the adversary starts out untrained, all compared algorithms start out with a fairly high score. As the adversary gradually learns to infer and move towards the target POI, MATD3 and MADDPG demonstrate a gradual decline in performance. However, MERL and EA are able to hold their performance by concocting effective counter-strategies in deceiving the adversary. EA reaches the same performance as MERL but is slower to learn.

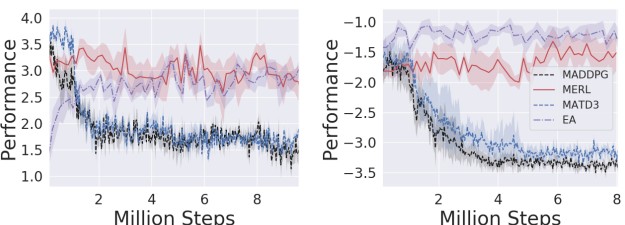

Figure 5: Performance on Physical Deception (left) and Keep-Away (right)

**Keep-Away:** Figure 5 (right) show the comparative performance in Keep-Away. Similar to Physical Deception, MERL and EA are able to hold performance by attaining good counter-measures against

the adversary while MATD3 and MADDPG fail to do so. However, EA slightly outperforms MERL on this task.

**Rover Domain:** Figure 6 shows the comparative performance of MERL, MADDPG, MATD3, and EA tested in the rover domain with coupling factors $1, 3$ and $7$. In order to benchmark against the proxy reward functions that use scalarized linear combinations, we test MADDPG and MATD3 with two variations of reward functions. *Global* represents the scenario where only the sparse team reward is used. *Mixed* represents the scenario where a linear combination of the team-reward and agent-specific reward is used. Each reward is normalized before being combined. A weighing coefficient of 10 is used to amplify the team-reward's influence in order to counter its sparsity. The weighing coefficient was tuned using a grid search (more details in Figure 7).

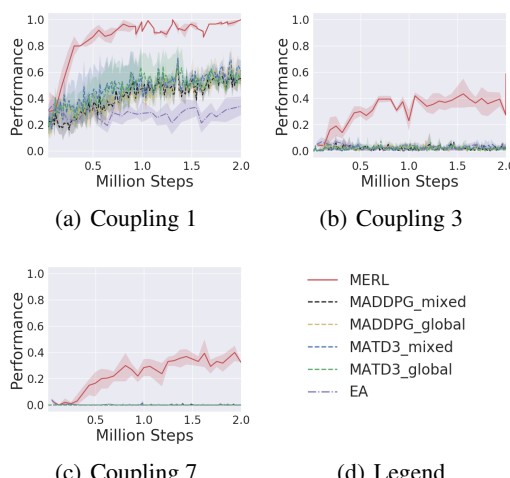

(a) Coupling 1       (b) Coupling 3

(c) Coupling 7       (d) Legend

Figure 6: Performance on the Rover Domain.

MERL significantly outperforms all baselines across all coupling requirements. The tested baselines clearly degrade quickly beyond a coupling of 3. The increasing coupling requirement is equivalent to increasing difficulty in joint-space exploration and entanglement in the team objective. However, it does not increase the size of the state-space, complexity of perception, or navigation. This indicates that the degradation in performance is strictly due to the increase in complexity of the team objective.

Notably, MERL is able to learn on coupling greater than $n = 6$ where methods without explicit reward shaping have been shown to fail entirely (Rahmattalabi et al., 2016). MERL successfully completes the task using the same set of information and coarse, unshaped reward functions as the other algorithms. The primary mechanism that enables this is MERL's split-level approach that allows it to leverage the agent-specific reward function to solve navigation and perception while concurrently using the team-reward function to learn team formation and effective coordination.

**Scalarization Coefficients for Mixed Rewards:** Figure 7 shows the performance of MATD3 in optimizing mixed rewards computed with different coefficients used to amplify the team-reward relative to the agent-reward. The results demonstrate that finding a good balance between these two rewards through linear scalarization is difficult, as all values tested fail to make any progress in the task. This is because a static scalarization cannot capture the dynamic properties of *which reward is important when* and instead leads to an ineffective proxy. In contrast, MERL is able to leverage both reward functions without the need to explicitly combine them either linearly or via more complex mixing functions.

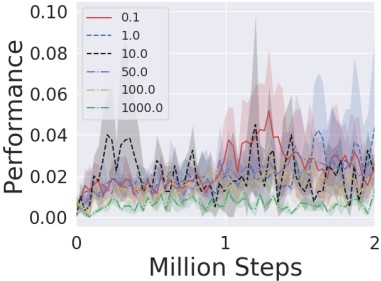

Figure 7: MATD3's performance for different scalarization coefficients

**Team Behaviors:** Figure 8 illustrates the trajectories generated for the Rover Domain with a coupling of $n = 3$. The trajectories for partially and fully trained MERL are shown in Figure 8 (a) and (b), respectively. During training, when MERL has not discovered team success (no POIs are successfully observed), MERL simply optimizes the agent-specific reward for each agent. This allows it to reach trajectories such as the ones shown in 8(a) where each agent learns to go towards a POI.

Since each agent explicitly aims to reach a POI, the probability 3 agents congregating to the same POI is higher compared to random undirected exploration by each agent without the dense agent-specific reward. Once this scenario is discovered, the team reward optimizer (EA) within MERL explicitly selects for agent policies that jointly lead to such team-forming behaviors. Eventually it succeeds

as shown in Figure 8(b). Here, team formation and collaborative pursuit of the POIs is immediately apparent. Two teams of 3 agents each form at the start of the episode. Further, the two teams also coordinate to pursue different POIs in order to maximize the team reward. While not perfect (the bottom POI is left unobserved), they do succeed in observing 3 out of the 4 POIs.

In contrast, MATD3-mixed fails to observe any POI. From the trajectories, it is apparent that the agents have successfully learned to perceive and navigate to reach POIs. However, they are unable to use this *skill* towards fulfilling the team objective. Instead each agent is rather split on the objective that it is optimizing. Some agents seem to be in sole pursuit of POIs without any regard for team formation or collaboration while others seem to exhibit random movements.

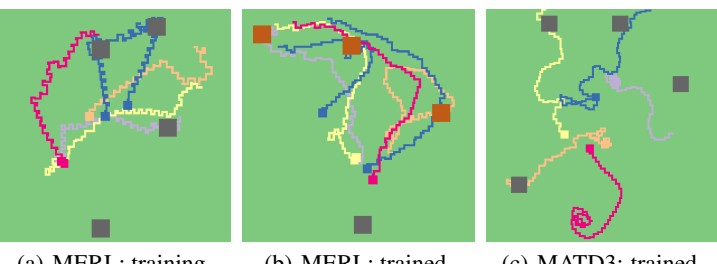

(a) MERL: training      (b) MERL: trained      (c) MATD3: trained

Figure 8: Agent trajectories for coupling = 3. Red/black squares are observed/unobserved POIs respectively

The primary reason for this is the mixed reward function that directly combines the agent-specific and team reward functions. Since the two reward functions have no guarantees of alignment across the state-space of the task, they invariably lead to learning these sub-optimal joint-behaviors that solve a certain form of scalarized mixed objective. In contrast, MERL by virtue of its bi-level optimization framework is able to leverage both reward functions without the need to explicitly combine them. This enables MERL to avoid these sub-optimal policies and solve the task without any reward shaping or manual tuning.

**Selection Rate:** We ran experiments tracking whether the policies migrated from the policy gradient learners to the evolutionary population were selected or discarded during the subsequent selection process (Figure 9). Note that the expected selection rate if chosen at random is 0.1 as 1 policy is migrated into a population of 10. In contrast, the selection rate for migrated policies is significantly higher across all benchmarks with the exception of Keep-Away. This is consistent with the performance results seen in Keep-Away where EA initially outperforms MERL. However, in general, these results indicate that MERL's integrative approach in combining the two optimization processes towards optimizing the team objective is crucial.

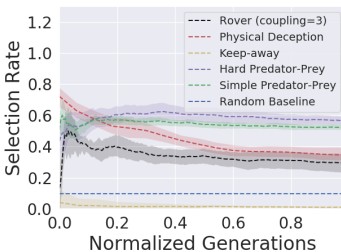

Figure 9: Selection rate for migrating policies

## 6 CONCLUSION

In this paper, we introduced MERL, a split-level algorithm that leverages both agent-specific and team objectives by combining gradient-based and gradient-free optimization. MERL achieves this by using a fast policy-gradient optimizer to exploit dense agent-specific rewards while concurrently leveraging neuroevolution to tackle the team-objective.

Results demonstrate that MERL significantly outperforms MADDPG, the state-of-the-art multiagent RL method, in a wide array of benchmarks. We also tested a modification of MADDPG to integrate TD3 - the state-of-the-art single-agent RL algorithm. These experiments demonstrated that the core improvements of MERL originate from its ability to leverage both team and agent-specific reward functions without the need to explicitly combine them. This differentiates MERL from other approaches like reward scalarization and reward shaping that either require extensive manual tuning or can detrimentally change the MDP (Ng et al., 1999) itself.

Future work will explore MERL for adversarial settings such as Pommerman (Resnick et al., 2018), StarCraft (Justesen and Risi, 2017; Vinyals et al., 2017) and RoboCup (Kitano et al., 1995; Liu et al., 2019). Further, extending MERL to general multi-reward settings such as is the case for multitask learning, is another promising area for future work.

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

## A  HYPERPARAMETERS DESCRIPTION

Table 1: Hyperparameters used for Predator-Prey, Keep-away and Physical Deception

| Hyperparameter | MERL | MATD3/MADDPG |
|---|---|---|
| Population size $k$ | 10 | N/A |
| Rollout size | 10 | 10 |
| Target weight $\tau$ | 0.01 | 0.01 |
| Actor Learning Rate | 0.01 | 0.01 |
| Critic Learning Rate | 0.01 | 0.01 |
| Discount Learning Rate $\gamma$ | 0.95 | 0.95 |
| Replay Buffer Size | $1e^6$ | $1e^6$ |
| Batch Size | 1024 | 1024 |
| Mutation Probability $mut_{prob}$ | 0.9 | N/A |
| Mutation Fraction $mut_{frac}$ | 0.1 | N/A |
| Mutation Strength $mut_{strength}$ | 0.1 | N/A |
| Super Mutation Probability $supermut_{prob}$ | 0.05 | N/A |
| Reset Mutation Probability $resetmut_{prob}$ | 0.05 | N/A |
| Number of elites $e$ | 4 | N/A |
| Exploration Policy | $\mathcal{N}(0, \sigma)$ | $\mathcal{N}(0, \sigma)$ |
| Exploration Noise $\sigma$ | 0.4 | 0.4 |
| Rollouts per fitness $\xi$ | 10 | N/A |
| Actor Neural Architecture | $[100, 100]$ | $[100, 100]$ |
| Critic Neural Architecture | $[100, 100]$ | $[300, 300]$ |
| TD3 Policy Noise variance | 0.2 | 0.2 |
| TD3 Policy Noise Clip | 0.5 | 0.5 |
| TD3 Policy Update Frequency | 2 | 2 |

Table 1 details the hyperparameters used for MERL, MATD3, and MADDPG in tackling predator-prey and cooperative navigation. The hyperparmaeters were inherited from Lowe et al. (2017) to match the original experiments for MADDPG and MATD3. The only exception to this was the use of hyperbolic tangent instead of Relu activation functions.

Table 2: Hyperparameters used for Rover Domain

| Hyperparameter | MERL | MATD3/MADDPG |
|---|---|---|
| Population size $k$ | 10 | N/A |
| Rollout size | 50 | 50 |
| Target weight $\tau$ | $1e^{-5}$ | $1e^{-5}$ |
| Actor Learning Rate | $5e^{-5}$ | $5e^{-5}$ |
| Critic Learning Rate | $1e^{-5}$ | $1e^{-5}$ |
| Discount Learning Rate $\gamma$ | 0.5 | 0.97 |
| Replay Buffer Size | $1e^5$ | $1e^5$ |
| Batch Size | 512 | 512 |
| Mutation Probability $mut_{prob}$ | 0.9 | N/A |
| Mutation Fraction $mut_{frac}$ | 0.1 | N/A |
| Mutation Strength $mut_{strength}$ | 0.1 | N/A |
| Super Mutation Probability $supermut_{prob}$ | 0.05 | N/A |
| Reset Mutation Probability $resetmut_{prob}$ | 0.05 | N/A |
| Number of elites $e$ | 4 | N/A |
| Exploration Policy | $\mathcal{N}(0, \sigma)$ | $\mathcal{N}(0, \sigma)$ |
| Exploration Noise $\sigma$ | 0.4 | 0.4 |
| Rollouts per fitness $\xi$ | 10 | N/A |
| Actor Neural Architecture | $[100, 100]$ | $[100, 100]$ |
| Critic Neural Architecture | $[100, 100]$ | $[300, 300]$ |
| TD3 Policy Noise variance | 0.2 | 0.2 |
| TD3 Policy Noise Clip | 0.5 | 0.5 |
| TD3 Policy Update Frequency | 2 | 2 |

Table 2 details the hyperparameters used for MERL, MATD3, and MADDPG in the rover domain. The hyperparameters themselves are defined below:

- **Optimizer = Adam**
  Adam optimizer was used to update both the actor and critic networks for all learners.

- **Population size** $k$
  This parameter controls the number of different actors (policies) that are present in the evolutionary population.

- **Rollout size**
  This parameter controls the number of rollout workers (each running an episode of the task) per generation.

  **Note:** The two parameters above (population size $k$ and rollout size) collectively modulates the proportion of exploration carried out through noise in the actor's *parameter* space and its *action* space.

- **Target weight** $\tau$
  This parameter controls the magnitude of the soft update between the actors and critic networks, and their target counterparts.

- **Actor Learning Rate**
  This parameter controls the learning rate of the actor network.

- **Critic Learning Rate**
  This parameter controls the learning rate of the critic network.

- **Discount Rate**
  This parameter controls the discount rate used to compute the return optimized by policy gradient.

- **Replay Buffer Size**
  This parameter controls the size of the replay buffer. After the buffer is filled, the oldest experiences are deleted in order to make room for new ones.

- **Batch Size**
  This parameters controls the batch size used to compute the gradients.

- **Actor Activation Function**
  Hyperbolic tangent was used as the activation function.

- **Critic Activation Function**
  Hyperbolic tangent was used as the activation function.

- **Number of Elites**
  This parameter controls the fraction of the population that are categorized as elites. Since an elite individual (actor) is shielded from the mutation step and preserved as it is, the elite fraction modulates the degree of exploration/exploitation within the evolutionary population.

- **Mutation Probability**
  This parameter represents the probability that an actor goes through a mutation operation between generation.

- **Mutation Fraction**
  This parameter controls the fraction of the weights in a chosen actor (neural network) that are mutated, once the actor is chosen for mutation.

- **Mutation Strength**
  This parameter controls the standard deviation of the Gaussian operation that comprises mutation.

- **Super Mutation Probability**
  This parameter controls the probability that a super mutation (larger mutation) happens in place of a standard mutation.

- **Reset Mutation Probability**
  This parameter controls the probability a neural weight is instead reset between $\mathcal{N}(0,1)$ rather than being mutated.

- **Exploration Noise**
  This parameter controls the standard deviation of the Gaussian operation that comprise the noise added to the actor's actions during exploration by the learners (learner roll-outs).

- **TD3 Policy Noise Variance**
  This parameter controls the standard deviation of the Gaussian operation that comprise the noise added to the policy output before applying the Bellman backup. This is often referred to as the magnitude of policy smoothing in TD3.

- **TD3 Policy Noise Clip**
  This parameter controls the maximum norm of the policy noise used to smooth the policy.

- **TD3 Policy Update Frequency**
  This parameter controls the number of critic updates per policy update in TD3.

## B  ROLLOUT METHODOLOGY

Algorithm 2 describes an episode of rollout under MERL detailing the connections between the local reward, global reward, and the associated replay buffer.

**Algorithm 2** Function Rollout

1: **procedure** ROLLOUT($\pi$, $\mathcal{R}$, noise, $\xi$)
2:     $fitness = 0$
3:     **for** j = 1:$\xi$ **do**
4:         Reset environment and get initial joint state $js$
5:         **while** env is not done **do**
6:             Initialize an empty list of joint action $ja = []$
7:             **for** Each agent (actor head) $\pi^k \in \pi$ and $s_k$ $in$ $js$ **do**
8:                 $ja \Leftarrow ja \cup \pi^k(s_k|\theta^{\pi^k}) + noise_t$
9:             **end for**
10:             Execute $ja$ and observe joint local reward $jl$, global reward $g$ and joint next state $js'$
11:             **for** Each Replay Buffer $\mathcal{R}_k \in \mathcal{R}$ and $s_k, a_k, l_k, s'_k$ in $js, ja, jl, js'$ **do**
12:                 Append transition $(s_k, a_k, l_k, s'_k)$ to $R_k$
13:             **end for**
14:             $js = js'$
15:             **if** env is done: **then**
16:                 $fitness \leftarrow g$
17:             **end if**
18:         **end while**
19:     **end for**
20:     Return $\frac{fitness}{\xi}$, $\mathcal{R}$
21: **end procedure**

## C    EVOLUTIONARY ALGORITHM POPULATION RUNS

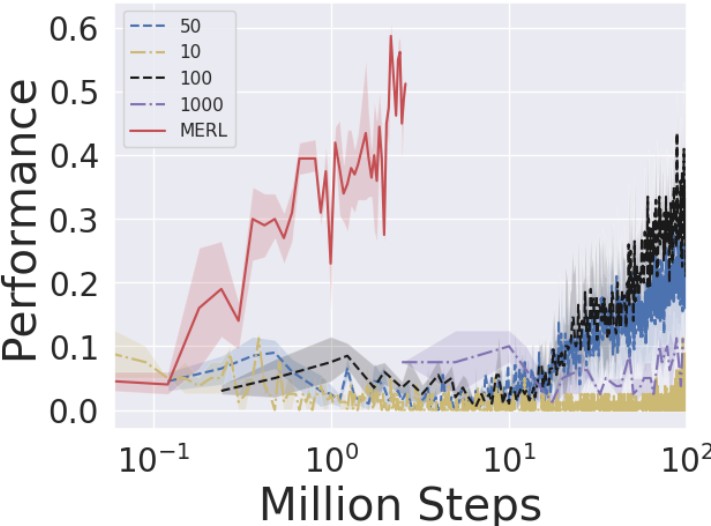

Figure 10: Evolutionary Algorithm Population size sweep on the rover domain with a coupling of 3. MERL was run for 2-million steps while the other EA runs were ran for 100-million steps.

Figure 10 compares EA with varying population sizes in the rover domain with a coupling of 3. Among the EA runs, a population size of 100 yields the best results converging to 0.3 in 100-millions frames. MERL (red) on the other hand is ran for 2-million frames and converges to 0.48. This is due to MERL's ability to leverage gradient descent from its policy gradient components that lead to significantly faster learning performance.

## D    EVOLUTIONARY STRATEGIES (ES)

### D.1    ES POPULATION SWEEP

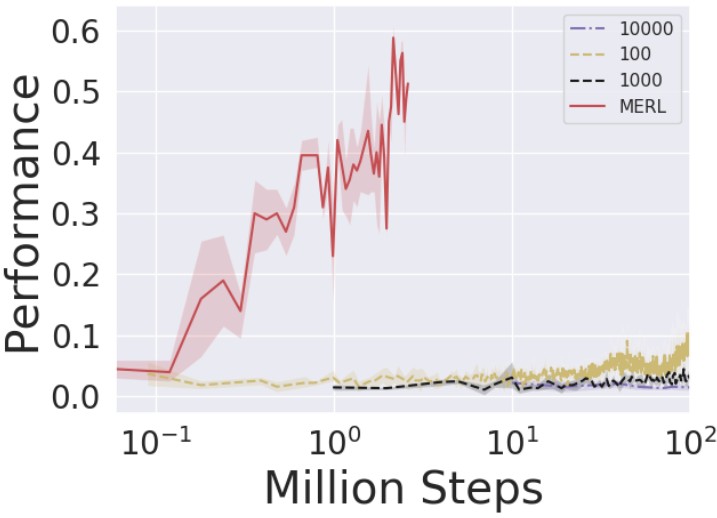

Figure 11: Evolutionary Strategies population size sweep on the rover domain with a coupling of 3

Figure 11 compares ES with varying population sizes in the rover domain with a coupling of 3. Sigma for all ES runs are set at 0.1. Among the ES runs, a population size of 100 yields the best results converging to 0.1 in 100-millions frames. MERL (red) on the other hand is ran for 2-million frames and converges to 0.48.

### D.2 ES SIGMA SWEEP

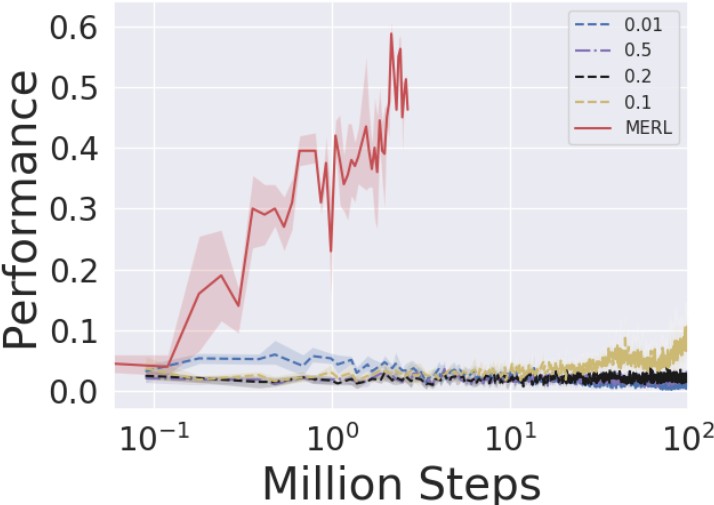

Figure 12: Evolutionary Strategies Noise magnitude (sigma) sweep on the rover domain with a coupling of 3

Figure 12 compares ES with varying variance of noises (sigma) that control the magnitude of each perturbation. The experiments are conducted in the rover domain with a coupling of 3 with a population size of 100. Among the ES runs, a sigma of 0.1 yields the best results converging to 0.1 in 100-millions frames. MERL (red) on the other hand is ran for 2-million frames and converges to 0.48.

## E PREDATOR-PREY WITH 3 PREY

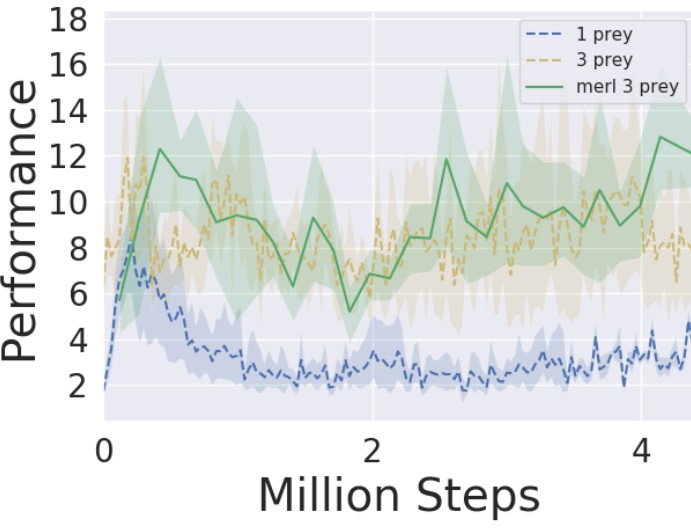

Figure 13: Predator-prey with varying numbers of prey. Prey are 30% faster than the predators

Figure 13 shows the results of running MATD3 with varying number of prey in the predator-prey domain. The experiments are ongoing.

