# OpenReview forum: "Evolutionary Reinforcement Learning for Sample-Efficient Multiagent Coordination"
_ICLR.cc/2020/Conference — Reject_

### Official Review · AnonReviewer2 · 2019-10-23
**Official Blind Review #2**

**Rating:** 6

**Review:**

This paper proposes an algorithm to learn coordination strategies for multi-agent reinforcement learning. It combines gradient-based optimization (Actor-critic) with Neuroevolution (genetic algorithms style). Specifically, Actor-critic is used to train an ensemble of agents (referred to as “team”) using a manually designed agent-specific reward. Coordination within a team is then learned with Neuroevolution. The overall design accommodates sharing of data between Actor-critic and Neuroevolution, and migration of policies. Evaluation is done using the multi-particle environments (Lowe et. al. 2017) and a Rover domain task.


I have the following questions:

1.	Comparison with PBT-MARL (Liu et al, 2019): PBT-MARL also proposed using gradient-methods (Retrace-SVG0) for learning with dense, shaped (local) rewards, and then using Neuroevolution to optimize the agents for coordination. I don’t think the description in Section 2 characterizes PBT-MARL well enough – reading that gives the impression that it only evolves the scalarization coefficients. PBT-MARL combines rewards with different discount factors (yielding it much more representation power than simple scalarization), and the discount factors are also evolved. Furthermore, the agent network weights are evolved based on team-reward, to learn coordination strategies.

At a qualitative level, this paper seems to be using a similar approach, albeit the specifics of Neuroevolution and policy-gradients are different. I would like the authors to shed light on the scenarios where they argue their method holds inherent advantage(s) compared to PBT-MARL.

2.	If I understand correctly, the individual agents in the team trained with DDPG would converge to similar policies – this is because each individual is trained with the same (agent-specific) reward, all agents in team use the same shared critic Q, and there is no flow of team from Neuroevolution to DDPG phase. This in itself is not a problem, because MADDPG should do the same if all agents are trained with the same team-reward function. The issue is that there are crucial differences in the architectures -- while the MADDPG paper had a separate Q network and policy network for each agent, MERL shares policy parameters (lower layers) between agents and uses a single, shared Q network. Have the authors done ablations with more aligned architectures, so that the improvements due to the main algorithmic contributions can be clearer?

3.	The notation defined in the background section is “s” for completed state (all agents) and “o” for individual observations. In the for-loop in Algorithm 1, are correct variables being used at all places, e.g. for pi, Q? In other words, which functions depend on the complete state of all agents, and which are decentralized? Additionally, should lines 23/24 be inside the for-loop?

4.	Experiments – In section 4, could the authors provide the values of N, L, K used in different environments, as applicable? The MADDPG paper claims good performance for multi-particle environments, which contradicts Figure 4 and 5. For example, MADDPG claims about 16 touches per episode with 30% faster prey, but Figure 4 has it converging to about 5. This makes me wonder if the parameters (N, L, K) are different in the two evaluations. Also, physical-deception task performance is reported in success% in the MADDPG paper, but the authors use a different distance metric. A standardized way of comparing performance would help the community.

5.	In MADDPG paper, an ensemble-based version is reported to perform much better. Since MERL is an ensemble method, is that not a more direct comparison?

Minor points:

1.	In environments (expect Rover), did the authors observe any benefits from adding the agent-specific reward to supplement the team-rewards (i.e. mixed reward setting) for the baselines?

2.	In Figure 8, why is the right-most POI lit when the red trajectory is far away from it? If the authors could provide a video of the MERL-trained policy for this domain, it’d be really cool.



----------------------Post-rebuttal update----------------------


I am satisfied with the author response and changes to the paper. I have increased my rating accordingly.

**Experience Assessment:**

I have read many papers in this area.

**Review Assessment: Checking Correctness Of Derivations And Theory:**

I assessed the sensibility of the derivations and theory.

**Review Assessment: Checking Correctness Of Experiments:**

I assessed the sensibility of the experiments.

**Review Assessment: Thoroughness In Paper Reading:**

I read the paper thoroughly.

---

> ### Author Response · Authors · 2019-11-11
> **Response 2**
>
> --->> "Parameter values of N, L, K and comparisons with the MA-DDPG paper results"
>
> For the rover-domain, N = 2*coupling so that there were enough agents to form two concurrent teams. L = 0 and K = 4 POIs.
>
> For the multiagent-envs, we inherited the values of N, L, K from the multiagent-env codebase that came with the MADDPG paper: https://github.com/openai/multiagent-particle-envs/tree/master/multiagent/scenarios.
>
> Upon a second review of the MADDPG paper, we realize that some of the default values in the codebase are different from those reported in the MADDPG paper.
>
> For example, predator-prey has L=2 in their codebase - which is also what we used for our paper. However, the results in their paper use L=3 (Table #3 in Appendix).
>
> Similarly, the number of preys we used for our benchmark is 1 - consistent with their codebase. Their paper does not reveal how many preys were used for their benchmark resulting in the 16 touches per episode - they simply mention 3 agents without clarifying if these are predator or prey agents. Since they report ~3x the number of touches (16 vs 5), a strong possibility is that they meant 3 agents of each type.
>
> We are performing one experiment that uses 3 predators and 3 preys for the predator-prey benchmark in the MADDPG setting (specifically, with MATD3 since TD3 is an improvement over DDPG). We do observe a significantly increased number of touches per episode although it is in the early stage of training. At the 4.3M samples stage, the 3-prey setting achieves about ~3x the number of touches as the 1-prey setting. Thus, it is conceivable that at ~10M samples (the convergence time for the results in our paper), the 3-prey setting will converge closer to 16 touches compared to 5 touches for the 1-prey setting. We report the latest available results in Appendix E in our revised manuscript.
>
> Having said that, the relative performance should not be affected for the results reported in our paper as all compared baselines were trained and tested in the same environment (i.e. 1 prey).
>
> --->> "In MADDPG paper, an ensemble-based version is reported to perform much better. Since MERL is an ensemble method, is that not a more direct comparison?"
>
> MADDPG-ensemble maintains multiple sub-policies for each agent. Each sub-policy also has its own individual buffer and learns independently from the other policies.
>
> While MERL does use a population of policies for each agent, they compete against each other and are constantly mixed using crossover-operations. Each agent (n'th team member) in MERL also maintains the same replay buffer and only has one policy/critic that uses this buffer to learn. Thus, MERL does not really use an ensembling method in the way MADDPG-ensemble does.
>
> Thus, comparing MERL to MADDPG-ensemble might not be the most direct comparison. In order to level the playing field for MERL, one would need to extend each agent in MERL to use an ensemble similar to MADDPG-ensemble. We expect MERL to also gain performance from such ensembling. We defer this to future work as it does not change the central message of the paper.
>
> --->> "In environments (expect Rover), did the authors observe any benefits from adding the agent-specific reward to supplement the team-rewards (i.e. mixed reward setting) for the baselines?"
>
> The environments other than Rover already provide mixed rewards as a baseline implementation for MADDPG and MATD3. We did not change these settings - thus the results reported for these environments only use the baseline mixed rewards.
>
> For neuroevolution, the results reported use only a team reward. However, we also tested this with added agent-specific rewards and observed consistently worse results. This was not reported in the paper in the interest of keeping the plots less cluttered and because we separately studied the impact of different mixing modes in the context of MADDPG and MATD3.
>
> --->> "In Figure 8, why is the right-most POI lit when the red trajectory is far away from it? If the authors could provide a video of the MERL-trained policy for this domain, it’d be really cool."
>
> The POI has an activation distance of 3 units around itself. If n rovers converge within this distance concurrently, they can activate the POI where n = coupling factor. The activation distance was not depicted in the visualization. In that specific figure, the red trajectory actually comes inside the required activation distance and helps satisfy the coupling-3 requirement. We will update the visualization to make this clearer. We also updated the anonymous github repo linked in the paper with videos demonstrating the different trajectories.

---

> ### Author Response · Authors · 2019-11-11
> **Response 1**
>
> Thank you for your insightful feedback.
>
> --->> "PBT-MARL combines rewards with different discount factors (yielding it much more representation power than simple scalarization), and the discount factors are also evolved."
>
> This is a very good point. Since PBT-MARL also evolves the discount rates associated with each reward and their mixing coefficients, they do have more representation power than the underlying simple linear mixing function. However, this setup still relies on the design of a mixing function to tackle the sparse team-reward. This is an important point of distinction with MERL that affects ease and flexibility of design as we further explain below.
>
> --->> "Furthermore, the agent network weights are evolved based on team-reward, to learn coordination strategies."
>
> In PBT-MARL, evolution only inherits policies (copies weights between agents) and does not evolve them directly (Algorithm 5 in Appendix B.5). Mutation and crossover only apply to the hyper-parameters (mixing coefficients, discount factor, etc) across agents and not to the weights themselves.
>
> This means that all the weight updates can only come from the underlying policy gradient algorithm (SVG0). This makes PBT-MARL’s performance entirely dependent on the policy gradient learner. Since SVG0 does not directly optimize the team-reward and only optimizes the mixture of agent-specific rewards, the emergence of coordination strategies strongly relies on the existence and discovery of a good mixing function.
>
> In contrast, MERL allows evolution to mutate/crossover the weights of the policy allowing it to directly search for coordination strategies that best maximize the team-reward. This is irrespective of whether a good mixing function exists - the split-level optimization of MERL alleviates the need to construct one.
>
> For illustration, consider the extreme scenario where the agent-specific rewards (shaping rewards) are entirely devoid of any useful information in optimizing the team-reward. In this scenario, MERL would automatically ignore the policy-gradient migrated policies and default to an EA to optimize the team-reward - thus allowing it to learn a usable policy.
>
> In this scenario, no choice of hyperparameters, discount factors or mixing coefficients would be able to align the information-absent local reward (shaping functions) towards optimizing the team-reward for SVG0 to use. In turn, since weight updates can only come from SVG0, PBT-MARL would not be able to learn a team strategy at all.
>
> --->> "The issue is that there are crucial differences in the architectures -- while the MADDPG paper had a separate Q network and policy network for each agent, MERL shares policy parameters (lower layers) between agents and uses a single, shared Q network. Have the authors done ablations with more aligned architectures, so that the improvements due to the main algorithmic contributions can be clearer?"
>
> We hypothesized that since the agents are homogeneous in their sensors and actuators, the perception and control modalities would be similar and perhaps would not need to be learned independently by each agent. This led to our design of sharing the policy parameters in the lower layers across all agents in a team. It is important to note that the MADDPG and MATD3 policies in our paper also use this parameter-sharing architecture. Thus all baselines use identical topologies in order to isolate the performance differences coming purely from the algorithmic differences.
>
> MADDPG implements separate but centralized critics in order to tackle non-stationarity. In contrast, MERL’s global optimizer (neuroevolution) is immune to non-stationarity as it directly evaluates the joint team performance independently for each rollout. Thus we designed MERL’s policy gradient algorithms with a shared and decentralized critic which is more compute and memory efficient.
>
> However, as suggested by the reviewer, we are implementing a version of MERL with separate but centralized critics as in MADDPG and will update the manuscript when the experiment completes. Since the centralized critic has access to more information, we expect this to only further improve MERL’s performance (at the cost of additional memory and compute) - and hence, not change the key conclusions of the paper.
>
> --->> "Notation errors"
>
> MERL is entirely decentralized, both during training and execution. All π and Q in MERL uses the observations o and never the complete state (s). We acknowledge our inconsistent use of s, o in the background and algorithm section which caused this confusion. We have updated the manuscript to remedy this confusion.
>
> Line 23 (soft update for target Q) should be inside the inner for-loop as we do a soft update after each gradient step. However, Line 24 is an error and should be outside the inner for-loop as migration happens every generation. We updated the manuscript with these corrections.

---

### Official Review · AnonReviewer3 · 2019-10-23
**Official Blind Review #3**

**Rating:** 8

**Review:**

This paper proposes to use a two-level optimization process to solve the challenge of optimizing the team reward and the agent's reward simultaneously, which are often not aligned. It applies the evolutionary algorithm to optimize the sparse team reward, while using RL (TD3) to optimize the agent's dense reward. In this way, there is no need to combine these two rewards into a scalar that often requires extensive manual tuning.

I vote for accepting this paper, because it tackles a practical problem in multi-agent learning. The presentation is clear, the algorithm is simple and sensible, the evaluation is thorough, and the results are better than the state-of-the-art.

The only question that I have is the sharing of replay-buffer for the same agent in different teams (Figure 2). For example, since the second agents in team1 and in team2 might learn to serve different roles in the task and may have completely different policies. I am not sure what is the purpose of this sharing. Considering the two extreme cases of replay buffer sharing, we could share all the data in a single replay buffer, or we could keep a separate replay buffer for each individual agent in each team, the paper chose the compromise between these two extremes. I wonder whether there is any theoretical or practical reasons to make this design choice. Is it important? I hope that the paper could have a deeper discussion if it is an important design decision.

----------------------Update after rebuttal----------------------

Thanks for the detailed response and the additional experiments. The response addressed my questions. Thus I will keep my original recommendation of acceptance.

**Experience Assessment:**

I have read many papers in this area.

**Review Assessment: Checking Correctness Of Derivations And Theory:**

I carefully checked the derivations and theory.

**Review Assessment: Checking Correctness Of Experiments:**

I assessed the sensibility of the experiments.

**Review Assessment: Thoroughness In Paper Reading:**

I read the paper at least twice and used my best judgement in assessing the paper.

---

> ### Author Response · Authors · 2019-11-11
> **Response**
>
> Thank you for your insightful feedback.
>
> This is a fair point. In the current setup for MERL, we constrain the n-th agents in each team to perform the same role as we share the replay buffer among them. As the reviewer stated, this is a middle ground between using a single buffer for all agents and a replay buffer for each agent in each team.
>
> In our observation, using a single buffer for all agents in every team leads to homogenization of the agents as all of them will lean towards the same behavior. On the other extreme case, each agent in each team having its own buffer greatly increases the memory and computational cost of learning while also curtailing information sharing across teams. Sharing the buffer for the n-th agent across all teams was the middle ground that led to good diversity (among team members) while remaining computationally and memory-wise tractable.

---

### Official Review · AnonReviewer1 · 2019-10-24
**Official Blind Review #1**

**Rating:** 1

**Review:**

This paper proposes an integration of neuroevolution and gradient-based learning for reinforcement learning applications. The evolutionary algorithm focuses on sparse reward and multiagent/team optimization, while the gradient-based learning is used to inject selectively improved genotypes in the population.
This work addresses a very hot topic, i.e. the integration of NE and DRL, and the proposed method offers the positive side of both without introducing major downsides. The presented results come from a relatively simple but useful multiagent benchmark which has broad adoption. The paper is well written, presents several contributions that can be extended and ported to other work, and the results are statistically significant.

There is one notable piece missing which forces me to bridle my enthusiasm: a discussion of the genotype and of its interpretation into the network phenotype. The form taken by the actual agent is not explicitly stated; following the adoption of TD3 I would expect a policy and two critics, for a grand total of three neural networks, but this remains unverified. And if each agent is composed of three neural networks, and each individual represents a team, does this mean that each genotype is a concatenation of three (flattened) weight matrices per each agent in the team? What is the actual genotype size? It sounds huge, I would expect to be at least several hundred weights; but then this would clash with the proposed minuscule population size of 10 (recent deep neuroevolution work from Uber uses populations THREE orders of magnitude larger). Has the population size been proportionated to the genotype dimensionality? Would it be possible to reference the widely adopted defaults of industry standard CMA-ES? Speaking of algorithms, where is the chosen EA implementation discussed? The overview seems to describe a textbook genetic algorithm, but that has been overtaken as state-of-the-art since decades, constituting a poor match for TD3.

Omitting such a chapter severely limits not only the reproducibility of the work but its full understanding. For example, does the EA have sufficient population size to contribute significantly to the process, or is it just performing as a fancy version of Random Weight Guessing? Could you actually quickly run RWG with direct policy search (rather than random action selection) to establish the effective complexity of the task? My final rating after rebuttal will vary wildly depending on the ability to cover such an important piece of information.

A few minor points, because I think that the paper appearance deserves to match the quality of the content:
- The images are consistently too small and hard to read. I understand the need to fit in the page limit by the deadline, but for the camera ready version it will be necessary to trim the text and rescale all images.
- The text is well written but often slowing down the pace for no added value, such as by dedicating a whole page to discussing a series of previously published environments.
- The hyperparameters of the evolutionary algorithm look completely unoptimized. I would expect a definite improvement in performance with minimal tuning.
- The "standard neuroevolutionary algorithm" from 2006 presented as baseline has not been state-of-the-art for over a decade. I would understand its usage as a baseline if that is indeed the underlying evolutionary setup, but otherwise I see no use for such a baseline.

-----------------------------------------------------------------------------------------------
# Update following the rebuttal phase
-----------------------------------------------------------------------------------------------

Thank you for your work and for the extended experimentation. I am confident the quality of the work is overall increased.

The core research question behind my original doubt however remains unaddressed: does the EC part of the algorithm sensibly support the gradient-descent part, or is the algorithm basically behaving as a (noisy) multi-agent TD3?
Such a contribution by itself would be undoubtedly important. Submitting it as a principled unification of EC and DL however would be more than a simple misnomer: it could mislead further research in what is an extremely promising area.

The scientific approach to clarify this point would be to design an experiment showcasing the performance of MARL using a range of sensible population sizes. To understand what "sensible" means in this context, I refer to a classic:
http://www.cmap.polytechnique.fr/~nikolaus.hansen/cec2005ipopcmaes.pdf
A lower bound for the population size with simple / unimodal fitness functions would be $4+floor(3*log(10'000)) = 31$. With such a complex, multimodal fitness though, no contribution from the EA can be expected (based on common practice in the EC field) without at least doubling or tripling that number. The upper bound does not need to be as high as with the recent Uber AI work (10k), but certainly showing the performance with a population of a few hundreds would be the minimum necessary to support your claim. A population size of 10 represents a proper lower bound for a genotype of up to 10 parameters; it is by no means within a reasonable range with your dimensionality of 10'000 parameters, and no researcher with experience in EC would expect anything but noise from such results -- with non-decreasing performance uniquely due to elitism.
The new runs in Appendice C only vary the population size for the ES algorithm, proposed as a baseline. No performance of MARL using a sensible population size is presented.

The fundamental claim is thereby unsustainable by current results. The idea is extremely intriguing and very promising, easily leading to supportive enthusiasm; it is my personal belief however that accepting this work in such a premature stage (and with an incorrect claim) could stunt further research in this direction.

[By the way, the reference Python CMA-ES implementation runs with tens of thousands of parameters and a population size of 60 in a few seconds per generation on a recent laptop: the claim of performance limitations as an excuse for not investigating a core claim suggests that more work would be better invested prior to acceptance.]


**Experience Assessment:**

I have published in this field for several years.

**Review Assessment: Checking Correctness Of Derivations And Theory:**

I carefully checked the derivations and theory.

**Review Assessment: Checking Correctness Of Experiments:**

I carefully checked the experiments.

**Review Assessment: Thoroughness In Paper Reading:**

I read the paper thoroughly.

---

> ### Author Response · Authors · 2019-11-11
> **Response**
>
> Thank you for your insightful feedback.
>
> We use a direct encoding setup where the genotype encodes the weights of the policy network. Each agent has one actor-critic set-up (specifically, one actor and two critics) that learns using policy gradients and, separately, an evolutionary population of k policy networks. The evolutionary population does not have any critic networks. During migration, the gradient-based policy is copied into the evolutionary population replacing one of the k policy networks there. Since neuroevolution does not use a critic, the critics are never migrated.
>
> The number of parameters varies by task but are in general within the range of tens of thousands for the policy.
>
> As the reviewer pointed out, a population size of 10 is tiny compared to the Uber AI papers. By itself, they are ill-equipped to train tens of thousands of policy parameters from scratch. This is where the policy migration from the policy gradient optimizer to the evolutionary population becomes crucial and is a critical contribution of this paper. We use the gradient-based learners to selectively inject improved genotypes into the evolutionary population. As the EA population acquires these “locally-trained” weights periodically, evolution can bootstrap its search using them. Thus the EA’s search space - and the required population size - is reduced greatly compared to methods that rely exclusively on evolution to train all of its weights.
>
> Further, EA uses the migrated networks as building blocks to select for alignment with the team objective - e.g., rovers picking POIs to go to that best maximize the global reward even if it is not the closest one (essential to team formation and spreading behavior). This mechanism allows us to achieve state-of-the-art performance without relying on a large population that is typical of purely evolutionary approaches.
>
> However, we acknowledge that increasing the population size would assuredly help learning (as corroborated in the Uber AI papers). However, this would cause the method to be extremely sample-expensive. For instance, for a rollout episode of 500 steps, one generation of evolution with a population size of 10 and 100 would cost 5,000 and 50,000 samples, respectively. This would be the difference between running 200 and 20 generations of evolution with a 2-million sample budget (in the case of the rover-domain). A key motivation for our paper is to learn coordination policies while also being overall sample-efficient. This is different from the focus on wall-clock time (leveraging EA/ES's parallelizability) in most contemporary approaches.
>
> We performed an additional experiment to evaluate the efficacy of different population sizes from 10-1,000 for the rover domain with a coupling factor of 3. All results were averaged over 5 runs. We added this result to Appendix C in our updated manuscript.
>
> We found the best EA performance at K=100 reaching ~0.3 in 100-million time steps. Compare this to MERL which reaches a performance of ~0.48 in only 2 million time steps. This demonstrates the efficacy of the guided evolution approach in our paper over purely evolutionary approaches.
>
> As the reviewer stated, we used a very simple genetic algorithm as our global optimizer without any parameter tuning. Tuning these parameters can only improve MERL’s performance. However, the key message in the paper is that MERL allows ease of design and does not need carefully tuned parameters.
>
> As proposed by the reviewer, we explored additional baselines. Since our parameter space was in the order of tens of thousands, CMA-ES was infeasible to run within the rebuttal period. Thus we chose Evolutionary-Strategies (ES) which has been successfully used in the recent Open AI work.
>
> We ran comparisons in the rover domain with a coupling factor of 3. All results were averaged over 5 runs.  The ES result is shown in Appendix D in our updated manuscript.
>
> In Appendix D.1, we vary the ES population size from 100-10,000. The top performance is ~0.1 at 100 million steps while MERL achieves ~0.48 in 2 million time steps.
>
> Apart from the population size, a key hyperparameter for ES is the variance of the perturbation factor (sigma). We run a parameter sweep for sigma and report results in Appendix D.2 - and did not find any major observable improvement.
>
> Further, we implemented RWG with a direct policy search using normal and uniform distribution to initialize the guesses for neural network weights. With 100 million samples, RWG led to a performance of ~0.009 and ~0.007 corresponding to normal and uniform weight initialization schemes, respectively. In comparison, MERL converged to 0.48 with a 2-million sample budget. The failure of RWG to learn can be attributed to the sparsity of the reward distribution - where 3 agents need to independently convene to a POI for a reward.
>
> We concede the points about writing style and formatting and will incorporate these suggestions in the final manuscript.

---

### Author Response · Authors · 2019-11-12
**Response to all reviewers**

We thank all the three reviewers for their in-depth analysis and questions on our paper. In our rebuttal below, we have addressed each reviewer's comments and questions.

Several of the responses required us to run additional experiments.  We have updated the manuscript with the results from these experiments. We added them to the Appendix section to highlight them and differentiate them from the existing plots in the original manuscript. We will integrate these results back into the main body of the paper in the camera-ready version.

Additionally, we have updated our anonymized Github repo (https://anonymous.4open.science/r/1590ffb0-aa6b-4838-9d59-ae20cdd8df11/) with all of the additional code that were used to generate the new results. These are available in the folder called "new_experiments". The repo also has a folder called "videos" containing a set of animations that demonstrate the different policies learnt by MERL vs MADDPG.

Specifically, the additional experiments we did are:
1. Ran EA for several different population sizes and compared them to MERL to address Reviewer 1's questions about the role of population sizes (Appendix C). This experiment demonstrated that MERL significantly outperforms purely evolutionary approaches for the several different population sizes explored.

2. Ran a version of ES (Evolutionary Strategies) as described in the Open AI Paper (https://arxiv.org/abs/1703.03864) to address Reviewer 1's question on more modern EA benchmarks. For this experiment, we investigated the role of population size (Appendix D.1) as well as hyper-parameter tuning (Appendix D.2).

3. Implemented two versions of RWG in response to Reviewer 1's suggestion. As we report below, RWG was unable to learn a policy even when trained up to 100M time steps.

4. Ran an experiment on the Predator-Prey benchmark showing the difference in performance between MADDPG for a setting with 3 preys vs 1 prey (Appendix E). This experiment was to address Reviewer 2's questions on apparent discrepancies in MADDPG's performance as reported in their paper vs ours. Based on this experiment (still training), it appears that the gap might be due to our use of 1 prey vs MADDPG's use of 3 preys (although they do not report this clearly in their paper).

5. Following reviewer 3's suggestion, we created an animation showing the trajectories of different agents as they team up to solve the Rover domain problem for a coupling of 3. The animation is rather rudimentary but clearly demonstrates MERL's well-coordinated strategies that successfully cover 3 out of 4 PoIs after 2M time-steps of training - while MADDPG is unable to form a coherent strategy for the same amount of training samples.

---

### Decision · Program_Chairs · 2019-12-19

**Decision:**

Reject

**Comment:**

This work has a lot of promise; however, the author response was not sufficient to address the concerns expressed by reviewer 1, leading to an aggregate rating that is just not sufficient to justify an acceptance recommendation. The AC recommends rejection.